# A Pilot COVID-19 Surveillance Program at the Zendrini Center in Milan (Italy) for Unaccompanied Foreign Minors

**DOI:** 10.3390/children9101485

**Published:** 2022-09-28

**Authors:** Stefano Tambuzzi, Marco Cummaudo, Lidia Maggioni, Stefania Tritella, Barbara Lucchesi, Paola Montedoro, Immacolata Agostinelli, Sofia Trezzi, Antonella Maria Costantino, Rossana Mazzoni, Michela Marognoli, Pasquale Poppa, Danilo De Angelis, Cristina Cattaneo

**Affiliations:** 1LABANOF (Laboratorio di Antropologia e Odontologia Forense), Institute of Forensic Medicine, Department of Biomedical Sciences for Health, University of Milan, Mangiagalli 37 Street, 20133 Milan, Italy; 2Unit of Radiology, IRCCS Policlinico San Donato, Rodolfo Morandi 30 Street, 20097 San Donato Milanese, Italy; 3Zendrini Centre, Municipality of Milan, Bernardino Zendrini 15 Street, 20147 Milan, Italy; 4U.O.N.P.I.A., Operative Unit of Neuropsychiatry of Childhood and Adolescence, IRCCS Fondazione Ca’ Granda, Ospedale Maggiore Policlinico, Festa del Perdono 7 Street, 20122 Milan, Italy

**Keywords:** unaccompanied foreign minors, migration, reception center, COVID-19, health surveillance program, nasopharyngeal antigenic swabs

## Abstract

During the COVID-19 pandemic, not only crowded refugee camps and immigration detention centers, but also receptions were places in which outbreaks occurred. To date there has been no report of the application of a COVID-19 surveillance system in reception centers for unaccompanied foreign minors only, who most of all deserve the utmost attention. Aware of this critical issue, we implemented a pilot COVID-19 surveillance program at the Zendrini center in Milan. It was started in September 2021 and was carried out for 4 months. Nasopharyngeal antigenic swabs were adopted. One day a week, two forensic physicians performed the first antigenic swab to minors who had just entered the center, or a monitoring swab after 15 days to those who were still hosted at the center. Operators were also swabbed for surveillance. A total of 80 subjects were enrolled and divided into 68 (72.5%) unaccompanied foreign minors and 22 (27.5%) operators. A total of 178 antigenic nasopharyngeal swabs were performed and tested negative. Regarding the monitoring activities, it was found that the minimum number of swabs per subject was 1 and the maximum number was 7, with an average value of 2.2 per individual. Having been able to confirm the absence of SARS-CoV-2 within the community represented a way to protect individual and collective health that could not have been pursued otherwise. Only inclusive approaches can allow communities and societies to respond more effectively to this crisis, and reduce the risk of future ones, intended as both upcoming COVID-19 waves and new infectious diseases.

## 1. Introduction

Since it was first described in December 2019, the severe acute respiratory syndrome coronavirus (SARS-CoV-2) causing coronavirus disease 2019 (COVID-19) has swept across the world affecting all countries and resulted in more than 6 million deaths [1]. Vulnerable and marginalized populations, such as ethnic minorities and migrant groups, have been found to be unduly affected. The latter have been disproportionately affected by mortality for COVID-19 worldwide, particularly in the United States and the United Kingdom [2,3,4]. Data on COVID-19’s effects on migrant morbidity and mortality are scarce; however, migrants residing in refugee camps, detention or reception facilities are at a heightened risk for the virus [2,5]. This statistic should not be overlooked because, according to the United Nations High Commissioner for Refugees (UNHCR), at the end of 2020, Europe alone was home to over 6.7 million migrants [6,7]. Moreover, it should never be overlooked that among migrants there are also minors (under the age of 18), whose percentage has been estimated at 38% by UNHCR [6,7]. Furthermore, minors are very often unaccompanied, and this condition of additional fragility requires a higher level of protection. With the aim of guaranteeing special protective measures for these minors, the European Union enacted the *Action Plan on Unaccompanied Minors 2010–2014* [6,8]. In response, some countries strengthened already existing protective measures for unaccompanied foreign minors and introduced new ones. In some cases, it has given further impulse to the improvement of reception centers where minors are hosted. This is the case of the multifunctional center “Centro Servizi Zendrini” in Milan, Italy. However, during the COVID-19 pandemic, not only crowded refugee camps and immigration detention centers [2,9], but also reception centers [2,5] were places in which outbreaks occurred. For this reason, the proper management of the COVID-19 pandemic among hosted migrants has become crucial to effectively contain and mitigate the outbreak, reduce the overall number of people affected and shorten the emergency situation [10]. However, the proper management of the early emergency is as important as the late one and its consequences [11]. Indeed, the occurrence of different pandemic waves over the months has made health surveillance systems imperative. Such surveillance programs have been implemented among the general population, but as very often happens the same precautions are not straightforwardly applied to certain categories of the population, such as migrants. To date, only one Italian experience of the surveillance of COVID-19 in migrant reception centers has been reported in the literature [11]. However, it was based only on the detection of body temperature and on the monitoring of influenza-like illness. On the other hand, routinely or experimental surveillance programs based on molecular or antigenic tests have been carried out in workplaces [12] and primary schools [13]. Nonetheless, to date there has been no report of the application of a COVID-19 surveillance system in reception centers for unaccompanied foreign minors only, who most of all deserve the utmost attention.

Aware of this critical issue, we implemented a pilot COVID-19 surveillance program at the Zendrini center in Milan, one of the cities in the Northern Italy most affected by migratory flows and most hit by the COVID-19 epidemic [14]. Therefore, we present the difficulties encountered, the solutions we adopted and discuss the importance of routinely implementing health surveillance programs in reception centers where migrants, especially unaccompanied minors, are hosted.

### The Setting

The multifunctional center “Centro Servizi Zendrini” in Milan (Italy) is open seven days a week, 24 h a day. The center operates both as a first level reception center for unaccompanied foreign minors only and a hub for the accomplishment of all the legal and bureaucratic procedures required by the Italian law. In the facility there is a multidisciplinary team comprising different professionals, such as educators, psychologists, intercultural mediators, forensic physicians and anthropologists. Since its opening in May 2019 to December 2021, the center hosted 608 minors between 12 and 18 years, with a mean age around 16. Of these minors, over 60% came from North Africa, mostly from Egypt, Tunisia and Morocco. Seventeen percent came from Asia, with Bangladesh and Pakistan as the most frequent countries of origin. Finally, 15% came from eastern European countries such as Albania and Kosovo. Other less frequently recorded native countries were Afghanistan, Somalia, Gambia, Senegal, Sud Sudan, Sudan, Nigeria and Mali. In over 80% of the cases, minors spontaneously arrived at the center, whereas in 13% they were brought by the police. Double/multiple rooms (twelve beds), outdoor/indoor common areas, canteen and a medical center are available. Residents have no restrictions for entry and exit and thus minors can spontaneously leave the facility. However, almost all the minors are hosted until the allocation of a place in the second reception communities. Therefore, minors may stay at the Zendrini center for a minimum of one week to a maximum of a few months.

Since the outbreak of the COVID-19 pandemic, minors can only enter the center after undergoing a negative nasopharyngeal molecular swab, which is carried out at the Policlinico Hospital in Milan. Moreover, a health promotion intervention was carried out to provide information on correct behaviors to avoid contagion and to raise awareness on the risks related to the epidemic and on the prevention tools. However, in view of the possibility of residents to freely enter and exit, as well as their interactions in common areas and young age, the situation was considered at risk. Therefore, we decided to implement a pilot surveillance program with the aim of intercepting and managing an eventually SARS-CoV-2 outbreak.

## 2. Materials and Methods

A surveillance program was started in September 2021 at the Zendrini center and conducted for 4 months. Two weeks prior to the project’s start, the center personnel received an email with the project’s design. The program was made available to all of the professionals working at the facility, even though the study was aimed at the minor residents of the center. As a result, anthropologists, forensic physicians, social workers, psychologists, educators and intercultural mediators were enrolled on a voluntary basis. Following receipt of the necessary information, all participants, or their legal representatives, signed a consent form according to the Declaration of Helsinki.

Nasopharyngeal antigenic swabs (LITUO^®^ Colloidal Gold nasal-swab COVID-19 Box antigen rapid tests) were adopted, as they were deemed to be the best compromise between the diagnostic efficacy and the need for minimal invasiveness. Indeed, the tests would be performed on patients in an ambulatory setting, many of whom would certainly have been minors. The kit we chose had a sensitivity of 94.53%, a specificity > 99% and an accuracy of 98.02%, thus fully meeting all the parameters dictated by the European Community [15].

Briefly, one day a week (regularly every Thursday afternoon) two forensic physicians performed the first antigenic swab to minors who had just entered the center, or a monitoring swab after 15 days to those who were still hosted at the center. Given the known rapid turnover of hosted minors, we had already predicted that not all minors would receive the same number of monitoring swabs. Operators were also swabbed for surveillance every 15 days. Similarly, both the forensic physicians swabbed each other periodically.

Prior to the performing of each swab, each patient’s body temperature was measured and a record of his or her health status over the past 14 days was filled in. In particular, influenza-like symptoms were assessed, such as fever >37.5 °C, dry cough, fatigue, sputum production, dyspnea, myalgia or arthralgia, sore throat, headache, vomiting, diarrhea, dysgeusia and anosmia, as well as their date of onset if present. Previous contacts with confirmed positive cases were also investigated. Finally, the patients’ vaccination status was assessed.

## 3. Results

A total of 80 subjects were enrolled, divided into 68 (72.5%) unaccompanied foreign minors hosted at the Zendrini center and 22 (27.5%) operators. Out of the total of all minors, only two subjects could not be subjected to nasopharyngeal antigenic swabs. In fact, both suffered from psychiatric disorders (severe phobia in one case and autism spectrum disorder in the other) that did not make them cooperative at all. However, a total of 178 antigenic nasopharyngeal swabs were performed, all of which were found to be valid (the control line was always displayed) and tested negative. Regarding the monitoring and surveillance activities, it was found that the minimum number of swabs per subject was 1 and the maximum number was 7, with an average value of 2.2 per individual. In detail:-Thirty-eight subjects (33 minors, 87%, and five operators, 13%) received only one swab;-Nineteen subjects (nine minors, 87%, and five operators, 47%) received two swabs;-Nine subjects (7 minors, 78%, and two operators, 23%) received three swabs;-Four subjects (all minors) received four swabs;-Four subjects (all minors) received five swabs;-Three subjects (one minor, 33%, and two operators, 67%) received six swabs;-Three subjects (all operators) received seven swabs.

Of the total 178 swabs, in 171 cases the medical history collected before the swab was negative for any symptoms in the previous 14 days, while in the remaining 8 cases it was positive. In seven cases, the operators (two forensic physicians and five educators) complained of recent sore throat, headache, fever, myalgia and/or asthenia; in one case, one operator reported being a contact of a family member who had tested positive for SARS-CoV-2. Only in one case some symptoms were reported by a minor, who complained of recent headache and sore throat.

Finally, we examined the vaccination status of the enrolled subjects, and we observed that all the 22 operators had already been vaccinated with at least the first dose. In contrast, out of the total number of minors, only six (approximately 10%) had received the first dose of the vaccine.

## 4. Discussion

The data processing revealed that an excellent participation of the subjects potentially enrollable in this pilot COVID-19 surveillance program occurred. Since the primary purpose was to monitor the spread of the SARS-CoV-2 virus in a semi-closed community (no restrictions for entry and exit) for unaccompanied foreign minors, such as the Zendrini center, a large participation was highly desirable.

In this regard, most of the operators voluntarily submitted to such periodic surveillance over time, as did almost all of the minors. However, it should be mentioned that in this population, despite the minimally invasive nature of the procedure, we found an initial condition of diffidence, which was overcome with the explanation of the procedure that would be carried out. Nonetheless, in rare circumstances the performing of nasopharyngeal antigenic swabs was not feasible due to the peculiar health conditions of two hosted minors. Indeed, both suffered from psychiatric disorders that did not make them cooperative at all being afraid of the nasopharyngeal swab. Therefore, we encountered critical issues that did not allow the COVID-19 surveillance on specific individuals. In light of this, from our perspective, an appropriate and inclusive health surveillance program could be based on the availability of different tests. The recent literature [13,16,17] has demonstrated that molecular and antigenic salivary swabs are also a proper tool for SARS-CoV-2. Therefore, having procedurally different tests could allow the needs of patients to be met and the broadest possible health surveillance program to be performed. However, on the whole, antigenic nasopharyngeal swabs proved easy to perform and were well-tolerated by almost all subjects, which is why from our perspective they were the optimal tool for our pilot COVID-19 surveillance program. Molecular nasopharyngeal swabs are undoubtedly the most accurate method for diagnosing SARS-CoV-2 infection [18], but because they are so much more invasive and require a lot of logistical support (including health care professionals), they are not appropriate for routine testing or surveillance.

In the present study, conducted during a COVID-19 wave in Italy, none of the 68 unaccompanied foreign minors and none of the 22 operators tested positive for SARS-CoV-2. We observed a perfect overlap between the molecular nasopharyngeal swab performed just before entering the center and the first surveillance antigenic nasopharyngeal swab. This finding is to be considered as further evidence of the appropriateness of the test we used. In addition, the nasal antigenic swab could be considered a valuable tool for the rapid assessment of a subject when the molecular swab is not available. Given the known rapid turnover of hosted minors, sometimes accelerated by voluntary leaving, we had already predicted that not all minors would receive the same number of monitoring swabs. However, in more than half of all tested subjects we were able to perform at least two swabs. In some cases, up to seven swabs were carried out on a single subject, and as expected, long-term monitoring on operators was easier than on minors. In addition, the possibility of performing periodic surveillance swabs also allowed us to bring diagnostic clarity to cases of subjects (mostly operators) who had complained of influenza-like symptoms in the previous 14 days. In such cases, a rapid diagnosis is crucial for the well-being of all people living and working at the reception center. In fact, it prevents symptomatic minors from being precautionarily isolated causing discomfort of both such individuals and all the other residents, as there are shared dormitories. In addition, a rapid diagnosis also allows the operators to keep working regularly and continue to be the main reference point for all the hosted unaccompanied minors. Indeed, during their stay, minors are assisted by operators who have the crucial task of helping these children and adolescents to face a socio-cultural reality which is often extremely different from the one they are used to [6]. This issue may be particularly challenging due to the linguistic, cultural and social differences linked to the heterogeneity of the migrants hosted [11]. For this reason, since their arrival to the center, minors are constantly supported by cultural mediators who, together with social workers, educators and psychologists, help ensure their mental health and well-being, always taking into account specific socio-cultural and religious needs. These helps were very valuable during the COVID-19 period and also during the health monitoring program. Indeed, the general psycho-physical well-being of the minors is always a priority for the center, and therefore, also during the health surveillance program, the social workers always made sure that there were never any conflicts with the mental health and other needs of the guests. In two cases, the examination was avoided due to psychiatric disorders that caused great discomfort in performing a nasopharyngeal swab. This also gave us the opportunity to look more closely at the consent forms obtained in the case of minors. We emphasize that all subjects were fully and adequately informed whether or not they wished to undergo nasopharyngeal swabbing before the swabs were taken. From this perspective, all minors at the center were able to freely exercise their right to self-determination and free choice whether or not to undergo the examination. Since the minors could not sign, the signature necessarily had to be provided by their legal representatives. In addition, the minors who were housed at the center were already familiar with the nasopharyngeal swab, having undergone it at the hospital prior to their admission to the center. On the whole, therefore, the health monitoring program did not trigger any psychophysical discomfort in the minors; on the contrary, it protected their well-being by meeting their wishes to continue sleeping in common areas and playing with friends. Eventually, if workplaces and schools can be closed temporarily while waiting scheduled re-openings [13], the same cannot be undertaken for reception centers, especially for those hosting unaccompanied foreign minors.

## 5. Conclusions

In summary, the COVID-19 pandemic has certainly brought attention to health inequities, but it has also created a chance to face the root causes of those disparities. Investing in implementing proper active surveillance programs and control strategies in reception centers to further improve their safety and protect the migrants’ well-being may definitely be a starting point. Migrants housed in refugee camps, receiving centers and detention facilities must be subject to national oversight and have access to medical care (which in the case of migrants must never be taken for granted as in many European countries automatic health care signifies already having some of administrative protection which may take months or years). In the case of unaccompanied foreign minors, these needs are also strongly emphasized in the Convention on the Rights of the Child, especially at the article 2, 3, 22, 24 [19]. Overall, the risk and cost of frequent samples could be mitigated by the potential for early detection of asymptomatic or presymptomatic migrants, hence minimizing the need for quarantine and its consequences.

Otherwise, excluding migrants from COVID-19 awareness and prevention activities, screening and testing undermines the effectiveness of relevant public health efforts [20]. Therefore, only inclusive approaches can allow communities and societies to respond more effectively to this crisis, and reduce the risk of future ones, including both upcoming COVID-19 waves and new infectious diseases. Considering all these aspects, monitoring the spread of SARS-CoV-2 virus and confirming its absence within a community such as the Zendrini center, represented a way to protect individual and collective health that could not have been pursued otherwise.

## Data Availability

The data that support the findings of this study are available from the corresponding author upon reasonable request.

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
