# Peer review of "A Pilot COVID-19 Surveillance Program at the Zendrini Center in Milan (Italy) for Unaccompanied Foreign Minors"

_children, 2022, doi:10.3390/children9101485_

Round 1
Reviewer 1 Report
Thank you for the opportunity review this timely piece of frontline work. I have a small number of issues that I would encourage the authors to consider:
1. Ethics. I note that the participants or their representatives signed informed consent forms. I would like to see more reflection on this, particularly regarding migrant/refugee children's rights and their capacity and agency to participate in - or decline to participate in - studies of this sort.
2. While I can see that a practical purpose has been served in delivering this work, and I can also see that there is value for practitioner communities in its reporting, I wonder if there are further questions raised during the study in relation to a possible conceptual or theoretical contribution? Specifically, while the report focuses on the efficacy or otherwise of the tools used (primarily the swabs), what was learned regarding accommodation of these children's mental health/well-being, socio-cultural, or religious needs in regard to the process that was introduced. If these issues were not considered, this should be reported as a serious limitation of the study.
Author Response
We thank you for your helpful and constructive criticism. In light of this, we have thoroughly revised the manuscript by responding to your suggestions. Welcoming the Reviewer's suggestion, we further discussed informed consent in the case of minors residing at the centre. We also clarified their free choice of whether or not to undergo the health surveillance program. We hope we have adequately met the Reviewer's requests.
Moreover, children’s mental health/well-being, and socio-cultural or religious needs are always a top priority for the centre. These aspects are constantly examined, which was further reinforced during the COVID-19 pandemic. We emphasize that all these aspects have therefore been taken into account. We thank the reviewer for allowing us to discuss this aspect further. We have implemented the discussion section accordingly.
Reviewer 2 Report
The author argues that there are little data on the impact of COVID-19 among migrants specifically. However migrant living in refugee camps area particularly vulnerable group. Therefore this research is in particularly relevant. It will be even relevant to include the Convention of the Rights of the Child, in particularly the articles 2,3, 22 and 24 in the argumentation of the relevance of surveillance programs at refugee camps in time of pandemics.
Author Response
Thank you very much for the valuable suggestion. We are pleased to have introduced the proposed reference. We completely agree with the reviewer that the Convention of the Rights of the Child very strongly supports the statements we have made.